# AIMing for Explainability in GNNs

## Abstract

As machine learning models become increasingly complex and are deployed in critical domains such as healthcare, finance, and autonomous systems, the need for effective explainability has grown. Graph Neural Networks (GNNs), which excel in processing graph-structured data, have seen significant advancements, but explainability for GNNs is still in its early stages. Existing approaches fall into two broad categories: post-hoc explainers and inherently interpretable models. Their evaluation is often limited to synthetic datasets for which ground truth explanations are available, or conducted with the assumption that each XAI method extracts explanations for a fixed network. We focus specifically on inherently interpretable GNNs (e.g., based on prototypes, graph kernels) which enable model-level explanations. For evaluation, these models claim inherent interpretability and only assess predictive accuracy, without applying concrete interpretability metrics. These evaluation practices fundamentally restrict the utility of any discussions regarding explainability. We propose a unified and comprehensive framework for measuring and evaluating explainability in GNNs that extends beyond synthetic datasets, ground-truth constraints, and rigid assumptions, while also supporting the development and refinement of models based on derived explanations. The framework involves measures of **A**ccuracy, **I**nstance-level explanations, and **M**odel-level explanations (AIM), inspired by the generic Co-12 conceptual properties of explanations quality (Nauta et al., 2023). We apply this framework to a suite of existing models, deriving ways to extract explanations from them and to highlight their strengths and weaknesses. Furthermore, based on this analysis using AIM, we develop a new model called XGKN that demonstrates improved explainability while performing on par with existing models. Our approach aims to advance the field of Explainable AI (XAI) for GNNs, offering more robust and practical solutions for understanding and interpreting complex models.

## 1 Introduction and Related Work

Explainability in machine learning is gaining importance, especially as models are applied in areas like healthcare (Ahmedt-Aristizabal et al., 2021), finance (Wang et al., 2022), and autonomous systems (Li et al., 2024). Meanwhile, Graph Neural Networks (GNNs) (Kipf & Welling, 2017) have emerged as powerful tools for handling graph data. While both fields are evolving rapidly, the exploration of Explainable AI (XAI) within the context of GNNs—and specifically for inherently interpretable GNNs—remains limited.

Most existing models serve as post-hoc explainers that aim to identify importance maps over input graphs (Ying et al., 2019a; Luo et al., 2020; Vu & Thai, 2020; Yuan et al., 2021; Magister et al., 2021; Lucic et al., 2022; Shin et al., 2022). Some leverage the Shapley-values approach from game theory (SHAP (Lundberg & Lee, 2017)) (Duval & Malliaros, 2021; Akkas & Azad, 2024). Thresholding these importance maps yields induced subgraphs that serve as explanations. While these methods offer some insights, they often lack reliability as they provide approximations rather than accurately reflecting the model's decision-making process. They can be inconsistent, oversimplify complex models, and do not guarantee trustworthiness, leading to potentially misleading explanations. Furthermore, since they do not influence the training phase, they fail to promote transparency from the start, making them less suitable for critical applications where interpretability is essential. Furthermore, prevailing practice in evaluating explainers relies on the availability of ground truth explanations, leading to experiments predominantly conducted on simple synthetic benchmarks (Ying et al., 2019b; Baldassarre & Azizpour, 2019; Luo et al., 2020; Azzolin et al., 2023; Lin et al., 2020).

For cases without available ground truths, metrics such as fidelity (Amara et al., 2024; Zheng et al., 2024; Longa et al., 2024), robustness (Bajaj et al., 2022), sufficiency, and necessity Tan et al. (2022); Chen et al. (2022) were proposed to assess how predictions change when input graphs are altered based on explanations from explainers. Since all explainers are applied to the same network, prediction differences can be compared directly across them. In contrast, our work focuses on inherently interpretable GNNs, where each method produces scores with different distributions, adding complexity to the evaluation process. Additionally, measures of explanation size sparsity are used in evaluations (Yu et al., 2022; Lucic et al., 2022). Agarwal et al. (2023) present an approach similar to ours by considering a broader range of metrics and examining changes in explanations. However, their work is also limited to post-hoc methods only.

Compared to the abundance of post-hoc methods, work on inherently interpretable GNNs is relatively limited Kakkad et al. (2023). Some methods use information constraints, such as attention mechanisms (Miao et al., 2022), while others employ structural constraints, which can additionally enable extraction of model-level explanations. Our focus is on the latter category, where the most prominent models include Prototypical Networks (Ragno et al., 2022; Zhang et al., 2021) and Graph Kernel Networks (GKNs) (Nikolentzos & Vazirgiannis, 2020; Cosmo et al., 2021; Feng et al., 2022). Both models employ unsupervised concept learning (Koh et al., 2020), where the model learns to identify concepts (trainable prototypes or graph filters) against which input graphs are compared. This comparison yields similarity scores that guide the prediction process. While investigation of these learned concepts aims to unravel the model's decision-making process, instance-level explanations that support their predictions are not available. Furthermore, these models claim explainability based solely on their design, without assessing any specific measures of explainability.

We contend that the evaluation practices for GNN explainers and interpretable GNNs is inadequate in the context of XAI. We argue that the advancement of XAI in GNNs is hampered by the lack of standardized metrics for evaluating explainability, making it difficult to determine which models are superior and under what circumstances, ultimately limiting their impact and practical applicability.

To effectively evaluate the XAI capabilities of various methods, it is essential to assess aspects such as correctness, consistency, and complexity of the explanations they provide. The Co-12 framework (Nauta et al., 2023) outlines 12 conceptual properties of explanation quality, aiming to standardize the evaluation process for XAI methods. This framework has since been adapted for application in computer vision models (Nauta & Seifert, 2023). We aim to expand this line of research specifically within the context of GNNs. Figure 1 illustrates examplar evaluation using our framework.

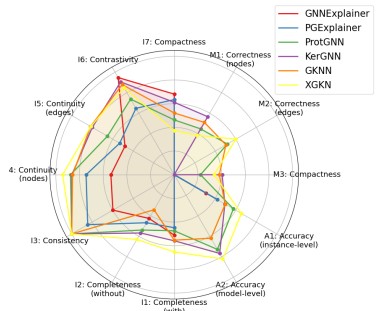

Figure 1: Evaluation of various XAI GNN methods over MUTAG data with proposed AIM metrics, including our XGKN model.

The primary objective of our research is to advance the field of Explainable AI for Graph Neural Networks. Recognizing the relative lack of exploration of evaluation metrics, we introduce a comprehensive set of metrics and a framework for evaluating XAI methods for GNNs, inspired by Co-12 properties. We derive ways to extract instance-level explanations from Prototypical Networks for graphs and Graph Kernel Networks, and asses these models in terms of their explainability. Building on our analysis of existing approaches and prior research on Graph Kernel Networks, we propose XGKN, a GKN model that demonstrates enhanced XAI capabilities.

Our contributions are as follows:

1. Proposal of AIM, a new evaluation framework for GNN explainability

   (a) Definition of AIM metrics for evaluating the XAI capabilities of GNN methods in terms of **A**ccuracy, **I**nstance-level explanations, and **M**odel-level explanations.
   (b) Development of a universal method for extracting instance-level explanations from Prototypical Networks for graphs and Graph Kernel Networks via SHAP propagation.
   (c) Comprehensive assessment of existing XAI approaches for GNNs.

2. Proposal of XGKN, a Graph Kernel Network with improved explainability.

Table 1: Summary of Co-12 properties covered by AIM.

| Property | Description |
| --- | --- |
| A1: Accuracy (instance-level) | Instance-level explanations should match with ground truths. |
| A2: Accuracy (model-level) | Model-level explanations should match with ground truths. |
| I1: Completness (with) | The graph and its explanation should both be classified into the same class. |
| I2: Completness (without) | Removing explanation from the graph should change the predicted class. |
| I3: Consistency | The explanation method should provide consistent results. |
| I4: Continuity (nodes) | Minor noise in node features should not significantly alter the explanation. |
| I5: Continuity (edges) | Minor noise in edges should not significantly alter the explanation. |
| I6: Contrastivity | Explanations for graphs of different classes should be distinguishable. |
| I7: Compactness | Explanations should be concise, small. |
| M1: Correctness (nodes) | If nodes in model-level explanation are altered, instance-level explanations should change. |
| M2: Correctness (edges) | If edges in model-level explanation are altered, instance-level explanations should change. |
| M3: Compactness | The set of all model-level explanations should be concise. |

## 2 AIM METRICS

Here, we outline the key properties of explainable GNN methods and propose a set of metrics for their evaluation. However, prior to that, it is crucial to define what constitutes an explanation in this context, considering current practices and their universality.

We assume that instance-level explanations take the form of induced subgraphs derived from the input graphs, whereas model-level explanations are graphs within the input space that are determined by identified concepts (prototypes in Prototypical Networks, or graph filters in Graph Kernel Networks). We choose to consider induced subgraphs (thresholded maps of importance over the input graphs) because the values within importance maps can vary in interpretation, range, and distribution across different models, which affects measures of similarity between maps.

Note that Prototypical Networks and Graph Kernel Networks can be defined as a composition of three functions $f = f_{pred} \circ f_{agg} \circ f_{sim}$, where $f_{sim}$ yields similarity scores between input subgraphs and concepts, $f_{agg}$ aggregates scores over subgraphs, and $f_{pred}$ does the final prediction. For Graph Kernel Networks, the input subgraphs correspond to the $k$-hop neighborhoods of each node, with the aggregation function usually being a summation. Prototypical Networks compare an encoded input graph against learned prototypes, while allowing subgraphs that contribute to the similarity scores to be identified. We assume that $f$ is a graph classification network, although this assumption can be omitted, and the formulas for the evaluation metrics can be easily adjusted.

### 2.1 PROPERTIES TO COVER

Co-12 (Nauta et al., 2023) is a set of conceptual properties, such as Compactness and Correctness, that are essential for a comprehensive assessment of explanation quality. Inspired by it, we propose AIM, a set of 12 metrics divided into 3 categories to assess: accuracy (A1-A2), instance-level explanations (I1-I7) and model-level explanations (M1-M3). These metrics cover 12 desired properties of GNN explanations that we describe in Table 1.

### 2.2 AIM EVALUATION FORMULAS

Let $\mathcal{G} \in \mathbf{D}$ be a graph from the dataset $\mathbf{D} \subset \mathbb{G}$. Let $f$ be a network that we want to explain—either an interpretable GNN or a GNN examined through post-hoc explainers. Let $h$ be an explainer, which for a graph $\mathcal{G}$, produces an induced subgraph $h(\mathcal{G}) \subset \mathcal{G}$ as an explanation, for the prediction $f(\mathcal{G}) = c$, where $c$ is the predicted class label. Let $\text{IoU}(\mathcal{G}_1, \mathcal{G}_2)$ denote the intersection over union of the node sets of induced subgraphs $\mathcal{G}_1, \mathcal{G}_2 \subset \mathcal{G}$. Let $\mathbb{I}$ represent the indicator function. Note that $h$ depends on $f$, and $h$ does not have to be deterministically defined—for example, when it has to be trained—while $f$ always returns the same result for the same input.

We now define the formulation for each of the AIM metrics using this notation. For specific details regarding perturbation methods and other hyperparameters, see Appendix A.1.

### 2.2.1 INSTANCE-LEVEL

**A1: Accuracy (instance-level)** Compares the instance-level explanation against the ground truth: $\text{IoU}\big(h(\mathcal{G}), \mathcal{E}\big)$, where $\mathcal{E}$ represents the ground truth explanation for the considered task, $\mathcal{E} \subset \mathcal{G}$.

**I1: Completeness (with)** Assesses class predictions for the explanation: $\mathbb{I}\big(f(h(\mathcal{G})) = c\big)$

**I2: Completeness (without)** Assesses class prediction for the induced subgraph that contains nodes not included in the explanation: $\mathbb{I}\big(f(\mathcal{G} \backslash h(\mathcal{G})) \neq c\big)$

**I3: Consistency** Measures consistency of the explainer: $\text{IoU}(h_1, h_2)$, where $h_1, h_2 \sim h(\mathcal{G})$.

**I4: Continuity (nodes)** Evaluates the differences in the explanation when the node features of the input graph are slightly modified: $\text{IoU}(h(\mathcal{G}), h(\mathcal{G}^{\text{nodes}}))$, where $\mathcal{G}^{\text{nodes}}$ is graph $\mathcal{G}$ with features altered in a few nodes.

**I5: Continuity (edges)** Evaluates the differences in the explanation when the edges of the input graph are slightly modified: $\text{IoU}(h(\mathcal{G}), h(\mathcal{G}^{\text{edges}}))$, where $\mathcal{G}^{\text{edges}}$ is graph $\mathcal{G}$ with a few altered edges.

**I6: Contrastivity** Assesses ability to distinguish between explanations for graphs classified into different classes: $\mathbb{I}\big(f^h(h(\mathcal{G})) = f(\mathcal{G})\big)$, where $f^h$ is a model trained on explanations $h(\mathcal{G})$ and predicted labels $f(\mathcal{G})$.

**I7: Compactness** Measures size of the explanation: $|h(\mathcal{G})|/|\mathcal{G}|$, where $|\cdot|$ measures graph size.

### 2.2.2 MODEL-LEVEL

Let $f(\cdot|\theta, \{\mathcal{H}_i\}_{i=1}^m)$ be an interpretable GNN network to be investigated (Prototypical Network or Graph Kernel Network), where $\mathcal{H}_1, ..., \mathcal{H}_m$ denote graphs that represent identified concepts (projected prototypes or graph filters) and $\theta$ represents the rest of the model's parameters, $m \in \mathbb{N}_+$. Note that $\mathcal{H}_1, ..., \mathcal{H}_m$ are model-level explanations.

**A2: Accuracy (model-level)** Compares model-level explanation against ground truths: $\frac{1}{l} \sum_{j=1}^l \min_{\mathcal{H}_i: i=1,...,m} \text{GED}(\mathcal{H}_i, \mathcal{E}_j)$, where $\mathcal{E}_1, ..., \mathcal{E}_l$ are graphs that are ground truth model-level explanations, $l \in \mathbb{N}_+$, and GED denotes normalized graph edit distance.

**M1: Correctness (nodes)** Evaluates the difference in the instance-level explanations, when node features in concepts are modified: $\sum_{\mathcal{G} \in \mathbf{D}} \text{IoU}(h(\mathcal{G}), h'(\mathcal{G}))$, where $h'$ is the explainer function of network $f(\cdot|\theta, \{\mathcal{H}_i^{\text{nodes}}\}_{i=1}^m)$, where $\mathcal{H}_i^{\text{nodes}}$ is $\mathcal{H}_i$ with modified node features.

**M2: Correctness (edges)** Evaluates the difference in the instance-level explanations, when node features in concepts are modified: $\sum_{\mathcal{G} \in \mathbf{D}} \text{IoU}(h(\mathcal{G}), h'(\mathcal{G}))$, where $h'$ is the explainer function of network $f(\cdot|\theta, \{\mathcal{H}_i^{\text{edges}}\}_{i=1}^m)$, where $\mathcal{H}_i^{\text{edges}}$ is $\mathcal{H}_i$ with altered edges.

**M3: Compactness** Assesses correlation between similarity scores with respect to different concepts: $1/(m(m-1)) \sum_{i=1}^m \sum_{j=i+1}^m \text{CORR}(\{s_i(\mathcal{G})\}_{\mathcal{G} \in \mathbf{D}}, \{s_j(\mathcal{G})\}_{\mathcal{G} \in \mathbf{D}})$, where $s_i(\mathcal{G}) = S_i$ for $S = f_{agg} \circ f_{sim}(\mathcal{G}|\theta, \{\mathcal{H}_i\}_{i=1}^m) \in \mathbb{R}^m$ and $i = 1, ..., m$.

Each proposed metric takes values in the range $[0, 1]$. Metrics I1-I6 should be maximized, while A1-A2, I7, M1-M3 should be minimized. Completeness is analogous to fidelity, sufficiency, or necessity as discussed in other works (Zheng et al., 2024; Tan et al., 2022; Chen et al., 2022), while continuity is comparable to robustness (Bajaj et al., 2022).

## 3 EVALUATION OF EXISTING XAI FOR GNNS USING AIM

### 3.1 EXTRACTING EXPLANATIONS FROM INTERPRETABLE GNNS

Post-hoc explainers for instance-level explanations provide methods for deriving explanations from existing GNNs in the form of importance maps over input graphs, whereas explanations from interpretable GNNs, such as Prototypical Networks for graph and Graph Kernel Networks, need to be extracted. First, we outline how the identified concepts are projected onto the input space to serve as model-level explanations. Next, we propose a SHAP-based approach for extracting instance-level explanations from concept-based interpretable GNNs.

As in the previous section, let $f = f_{pred} \circ f_{agg} \circ f_{sim}$ represent an interpretable GNN, where $f_{sim}$ yields similarity scores between input subgraphs and concepts, $f_{agg}$ aggregates scores over subgraphs, and $f_{pred}$ does the final prediction.

### 3.1.1 Model-level explanations

Existing interpretable GNN methods rely on trainable concepts to make predictions, which are regarded as model-level explanations. These concepts are either explicitly given in graph form (in Graph Kernel Networks) or as embeddings (in Prototypical Networks), which can be projected onto the input space to obtain their graph representations (Zhang et al., 2021). If graph filters in GKN are represented using continuous node features and a continuous adjacency matrix, they can be projected onto the input space by discretizing the adjacency matrix and identifying node features present in the dataset that show the highest similarity with respect to $f_{sim}$.

### 3.1.2 Instance-level explanations via SHAP propagation

For instance-level explanations, since no framework exists in prior work on interpretable GNNs, we define a method based on SHAP (Lundberg & Lee, 2017). While the described approach is tailored for graph classification tasks, it can be easily adapted to other use cases.

Let $\mathcal{G}$ be an input graph, let $m$ be the number of concepts identified by the network $f$, $m \in \mathbb{N}_+$. Let $S = f_{sim}(\mathcal{G}) \in \mathbb{R}^{n \times m}$, where $n$ denotes number of input subgraphs that are compared against concepts, $z = f_{agg}(S) \in \mathbb{R}^m$, and $p = f_{pred}(z) \in \mathbb{R}^c$, where $c$ is the number of classes, $c \in \mathbb{N}_+$. Let $\hat{p}$ be the logit predicted for the class that $\mathcal{G}$ will be classified as: $\hat{p} = \max_{i=1,\dots,c} p_i$.

First, we compute the SHAP values for the function $f_{pred}$, with input $z$ and prediction $\hat{p}$, yielding a set of values $\{\phi_i\}_{i=0}^m$, where $\sum_{i=0}^m \phi_i = \hat{p}$. Here, $\phi_0$ represents the expected value, while $\phi_i$ for $i = 1, \dots m$ corresponds to the importance of each respective concept. For simplicity, we assume that the aggregation function $f_{agg}$ is a summation over subgraphs, $z_i = \sum_{j=1}^n S_{ji}$. However, this can easily be adapted to accommodate other aggregation techniques. We define a map of importance over input subgraphs $w \in \mathbb{R}^n$ such that $w_j = \sum_{i=1}^m (\phi_i \cdot S_{ji})/z_i$, and hence $\phi_0 + \sum_{j=1}^n w_j = \hat{p}$.

For Prototypical Networks, input subgraphs are subgraphs that contribute to the similarity scores the most. Let $\mathcal{G}_i$ represent input subgraph that is associated with similarity score $z_i$, $i = 1, \dots, m$. We define importance of a node $v \in \mathcal{G}$ as $\psi(v) = \sum_i w_i$ for $i : v \in \mathcal{G}_i$, $i = 1, ..., m$. For Graph Kernel Networks, input subgraphs are defined as the neighborhoods of individual nodes. Assuming that nodes in $\mathcal{G}$ are ordered, let $v \in \mathcal{G}$ be the $i$-th node in graph $\mathcal{G}$, $i = 1, \dots, n$. Then $\mathcal{G}_i$ represents the subgraph centered around $v$. The importance of node $v$ is then defined as $\psi(v) = w_i$.

The final map of nodes importance is defined as $\text{softmax}(\{\psi(v)\}_{v \in \mathcal{G}})$. Using softmax normalizes the importance scores into a probability distribution, enabling clear comparison of each element's contribution. To identify the set of important nodes and, subsequently, the induced subgraph of $\mathcal{G}$ that serves as the explanation, we apply thresholding techniques.

## 3.2 Experiments

### 3.2.1 Setup

**Models** We evaluate different types of XAI models: 1) post-hoc explainers: GNNExplainer (Ying et al., 2019a) and PGExplainer(Luo et al., 2020), 2) Protypical Network: ProtGNN (Zhang et al., 2021), and 3) Graph Kernel Networks: KerGNN (Feng et al., 2022) and GKNN (Cosmo et al., 2021). Additionally, we train a GIN (Xu et al., 2019) for evaluation of post-hoc explainers.

**Datasets** We use 6 well-known datasets that contain: 1) synthetic graphs: BA2Motifs (Luo et al., 2020) and BAMultiShapesDataset (Azzolin et al., 2023), 2) molecular graphs: MUTAG (Debnath et al., 1991) and PROTEINS (Borgwardt et al., 2005), and 3) social graphs: IMDB-BINARY and IMDB-MULTI (Yanardag & Vishwanathan, 2015). Following common practices, we don't use node features in synthetic datasets, whereas in social datasets, node degree is used as the feature if model allows for it (all except GKNN). Table 2 summarizes statistics for the datasets.

Table 2: Dataset statistics summary.

| Dataset | # Graphs | Avg. # Nodes | # Classes |
|---|---|---|---|
| BA-2motif | 100 | 25 | 2 |
| BAMultiShapes | 1000 | 40 | 2 |
| MUTAG | 188 | 18 | 2 |
| PROTEINS | 1113 | 39 | 2 |
| IMDB-B | 1000 | 20 | 2 |
| IMDB-M | 1500 | 13 | 3 |

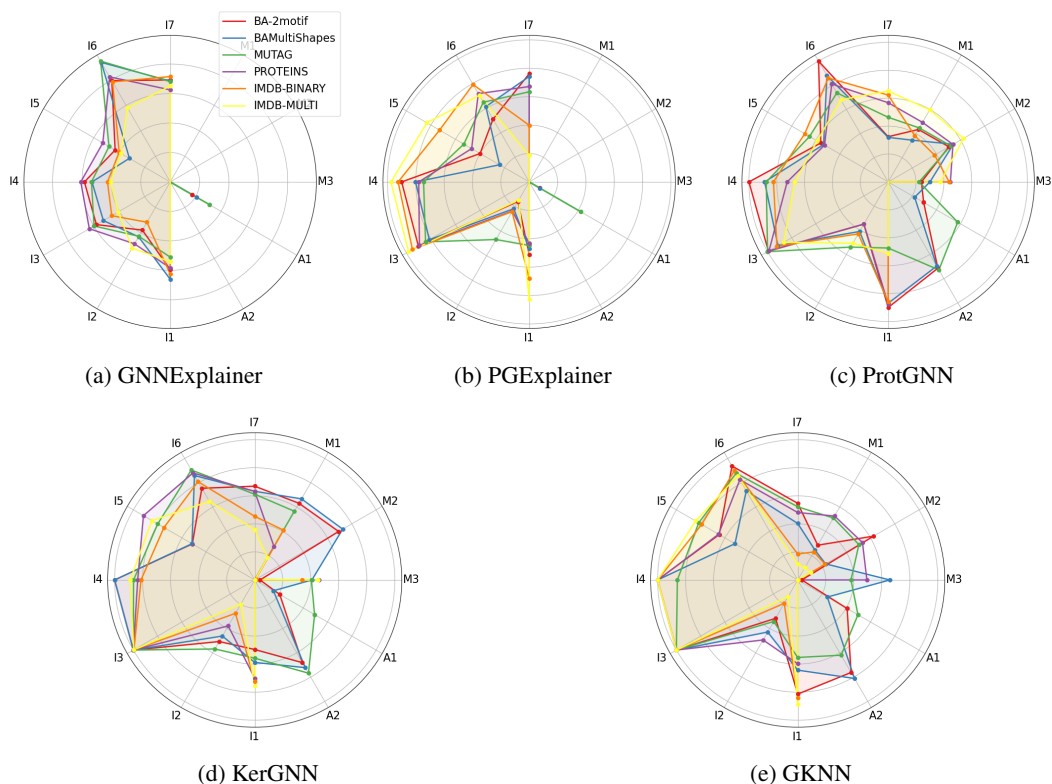

Figure 2: AIM metrics measured for post-hoc explainers (GNN-Explainer, PGE-Explainer), Prototypical Network (ProtGNN), and Graph Kernel Networks (KerGNN, GKNN). Note that the metrics have been oriented such that higher values indicate better performance.

**Metrics**   We evaluate prediction accuracy of each model as a reference point, and AIM matrices defined in Section 2.2. Each AIM metric has values in the range $[0, 1]$. For evaluation, minimization metrics are adjusted by using $1 - \gamma$, where $\gamma$ represents the metric value, ensuring that a higher score consistently indicates better performance across all metrics.

**Hyperparameters**   We select hyperparameters based on the authors' guidelines and optimize them for the best predictive accuracy. However, we observe that higher accuracy can sometimes result in lower XAI performance, as we notice in the case of KerGNN. For calculating SHAP, we use Deep SHAP (Lundberg & Lee, 2017). Hyperparameters specific to the calculation of AIM metrics are provided in the Appendix A.1.

**Thresholding**   To determine which nodes or edges should be included in the explanation based on maps of importance, we use elbow points as a thresholds. We evaluate node importance for all models, except for PGExplainer, which considers edge importance. Results for alternative thresholding techniques are in Appendix A.2.

### 3.2.2   RESULTS

Prediction accuracies of each model are shown in Table 3a, while Table 3b shows time needed to extract explanations using each method. Evaluation of AIM metrics is presented in Figure 2.

**GNNExplainer**   Based on Figure 4a, we observe that GNNExplainer does not provide consistent explanations (I3-I5), which may be attributed to the small size of the extracted explanations (I7). Refining the thresholding technique could improve this (see Appendix A.2). While the explanations are distinguishable between classes (I6), the model struggles to identify the most relevant parts of the graph for the task (I1-I2). Its similarity to ground truth is the weakest among all evaluated mod-

els. However, the method demonstrates robustness by maintaining consistent performance across datasets.

**PGExplainer** While PGExplainer, in Figure 4b, is a non-deterministic algorithm (has to be trained), it exhibits notable consistency (I3-I4), though it is understandably more sensitive to changes in edges, as they are central to its approach (I5). However, PGExplainer struggles to differentiate between relevant and irrelevant parts of the graph (I1-I2). Compared to other methods, its explanations lack clarity in distinguishing between classes (I6), which correlates with the size of the extracted subgraphs (I7).

**ProtGNN** ProtGNN's, in Figure 4c, limitations stem from its mechanism of sampling input subgraphs to generate prototype projections and identify the most relevant subgraphs. Since the sampling process heavily relies on edges, it becomes inconsistent when the input edges are altered (I5). While the prototypes are not explicitly in graph form and require projection onto the input space, ProtGNN still demonstrates a good level of explainability. However, the model's main drawback is its computational time (see Table 3b), largely due to the sampling strategy. Not only is the explanation extraction process slow, but the projection phase during training is also time-consuming, as it requires iterating over the dataset to identify subgraphs that best match the prototypes. This approach limits the model's scalability compared to others.

**KerGNN** KerGNN's issues, highlighted in Figure 4d, arise from its heavy reliance on node features rather than edges. While edges define the input subgraphs compared against graph filters, the model often performs best when the Random Walk Kernel—used as the kernel function—considers only paths of length 1. This essentially reduces the kernel function to a comparison of the node sets of graphs, largely ignoring edge structures. Additionally, KerGNN combines input node feature information with similarity scores for final predictions, which limits the model's ability to identify meaningful concepts, as it can infer much of the information just from node features (such as node degree in IMDB datasets).

**GKNN** As illustrated in Figure 4e, GKNN's explanations are consistent (I3-I5) and dependent on the learned concepts (M1-M2). Moreover, these concepts are both relevant (A1-A2) and concise (M3). We observe that the explanations tend to be larger (I7), which correlates with other metrics (I1-I2 and I6). This occurs because many subgraphs within a graph receive similar kernel responses from graph filters, making it more challenging to differentiate between relevant and irrelevant subgraphs, and consequently, individual nodes. The time required to extract explanations from GKNN is approximately 40% longer (see Table 3b) due to the use of graph kernels, such as the Weisfeiler-Lehman Graph Kernel, which cannot be executed on GPUs and involve slower operations compared to the Random Walk Kernel used in KerGNN. GKNNs employ non-differentiable graph kernels, and hence require training through a Discrete Randomized Descent strategy, which limits fast GPU computations and ultimately restricts their scalability.

Our observations suggest that KerGNNs do not effectively identify graph concepts. However, we acknowledge KerGNN's conceptual advantage over other methods, particularly in their scalability. In the following section, we introduce a new model built on the principles of KerGNN and other GKNs, designed to achieve a higher level of explainability by: 1) extracting more relevant concepts, and 2) simplifying the differentiation between relevant and irrelevant nodes in importance maps.

Table 3: Evaluation of models

(a) Accuracy for GNN classifiers.

| Dataset | GIN | ProtGNN | KerGNN | GKNN |
|---|---|---|---|---|
| BA-2motifs | 99.80 | 99.80 | 98.64 | 99.20 |
| BAMultiShapes | 95.10 | 87.00 | 90.30 | 93.90 |
| MUTAG | 84.74 | 89.47 | 80.05 | 82.63 |
| PROTEINS | 74.48 | 85.00 | 73.94 | 73.05 |
| IMDB-B | 76.00 | 71.78 | 71.10 | 66.00 |
| IMDB-M | 46.13 | 46.00 | 47.13 | 44.00 |

(b) Average runtime (s) for extracting explanations.

| Dataset | GNNExp. | PGExp. | ProtGNN | KerGNN | GKNN |
|---|---|---|---|---|---|
| BA-2motif | 0.014 | 0.029 | 44.948 | 0.033 | 0.043 |
| BAMultiShapes | 0.014 | 0.029 | 164.462 | 0.037 | 0.051 |
| MUTAG | 0.018 | 0.038 | 13.457 | 0.010 | 0.016 |
| PROTEINS | 0.012 | 0.027 | 43.497 | 0.038 | 0.058 |
| IMDB-B | 0.014 | 0.043 | 71.696 | 0.032 | 0.044 |
| IMDB-M | 0.011 | 0.039 | 10.223 | 0.066 | 0.081 |

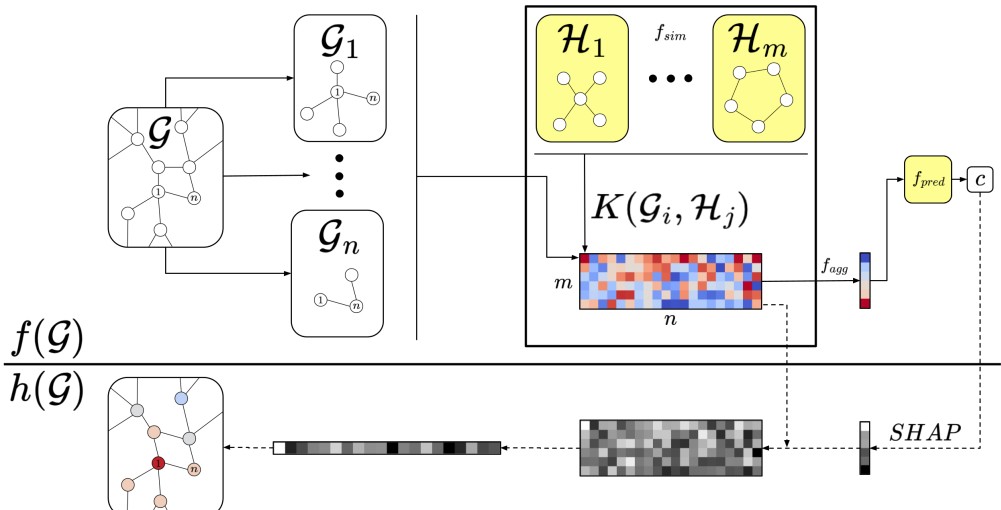

Figure 3: Overview of XGKN. Trainable components highlighted in yellow. Upper section illustrates the forward pass for prediction, and lower section demonstrates the extraction of explanations. Input graph $\mathcal{G}$ of size $n$ is processed as a set of node-centered subgraphs $\mathcal{G}_1, ..., \mathcal{G}_n$. Each subgraph is compared against graph kernels $\mathcal{H}_1, ..., \mathcal{H}_m$ using kernel function $K$, which yields similarity scores. These scores are then aggregated using $f_{\text{agg}}$ and passed to the predictor $f_{\text{pred}}$ which determines the class label $c$. For explanation extraction, SHAP values obtained for $f_{\text{pred}}$ are propagated back onto the input graph $\mathcal{G}$ by reversal of the aggregation of similarity scores.

## 4 XGKN

In this section, we introduce a new model that builds upon existing GKN principles and demonstrates enhanced explainability capabilities. Our primary objective is to refine the GKN model's ability to identify more relevant concepts and generate maps of node importance that offer a clearer distinction between relevant and irrelevant nodes compared to existing GKNs, particularly KerGNNs.

### 4.1 METHOD

Consistent with previous notations, we define the network XGKN as a composition of three functions $f_{pred} \circ f_{agg} \circ f_{sim}$. Function $f_{sim}$ extracts similarity scores (kernel responses), $f_{agg}$ aggregates them, and $f_{pred}$ produces final prediction.

Let $\mathcal{G}$ be an input graph of size $n$. Let $\mathcal{G}_v$ be a $k$-hop neighborhood of node $v$, $k \in \mathbb{N}_+$. Let $\mathcal{H}_1, ..., \mathcal{H}_m$ be the set of $m$ graph filters, $m \in \mathbb{N}_+$. Here, nodes in graphs are ordered. To simplify the notation, for $v \in \mathcal{G}$, $v$ denotes a node in $\mathcal{G}$ and also corresponds to its index in $\mathcal{G}$, $v = 1, \ldots, n$.

Function $f_{sim}$ represents the graph kernel module which extract kernel responses. In XGKN, we use Random Walk Kernel as it is computationally efficient and differentiable. It counts the number of walks that two graphs have in common. Let $\mathcal{G}, \mathcal{G}' \in \mathbb{G}$ be graphs. Since performing a random walk on the direct product graph $\mathcal{G}_\times = \mathcal{G} \times \mathcal{G}'$ is equivalent to performing the simultaneous random walks on graphs $\mathcal{G}$ and $\mathcal{G}'$, the $P$-step random walk kernel can be defined as

$$K_{RW}(\mathcal{G}, \mathcal{G}') = \sum_{p=0}^{P} K_{RW}^p(\mathcal{G}, \mathcal{G}') = \sum_{p=0}^{P} S^T A_\times^p S, \tag{1}$$

where $A_\times$ is an adjacency matrix of $\mathcal{G}_\times$, and $S = X X'^T$, where $X$ and $X'$ represents node features of $\mathcal{G}$ and $\mathcal{G}'$, respectively. $S_{ij}$ corresponds to similarity between $i$-th node in $\mathcal{G}$ and $j$-th node in $\mathcal{G}'$.

We want to associate kernel responses for $\mathcal{G}_v$ with the importance of node $v$, hence we define the kernel function $K : \mathbb{G} \times \mathbb{G} \to \mathbb{R}_+$ as a Random Walk Kernel in which only walks that start in $v$ are counted. To improve the fairness of the comparison of kernel responses from different graph filters,

we normalize feature embeddings of compared node features. For filter $\mathcal{H}_i$, $i = 1, \ldots m$, we obtain

$$K(\mathcal{G}_v, \mathcal{H}_i) = \sum_{p=0}^{|\mathcal{H}_i|} \sum_{\substack{u \in \mathcal{G}_v \\ v', u' \in \mathcal{H}_i}} (S^T A_\times^p S)_{(v,v'),(u,u')}, \quad (2)$$

where $A_\times$ is an adjacency matrix of $\mathcal{G}_v \times \mathcal{H}_i$, and $S = X_{\mathcal{G}_v} X_{\mathcal{H}_i}^T$, where $X_{\mathcal{G}_v}$ and $X_{\mathcal{H}_i}$ represent normalized (encoded) node features of $\mathcal{G}_v$ and $\mathcal{H}_i$, respectively. We define

$$f_{sim}(\mathcal{G}) = [K(\mathcal{G}_v, \mathcal{H}_i)]_{\substack{v=1,\ldots,n \\ i=1,\ldots,m}} \in \mathbb{R}_+^{n \times m}. \quad (3)$$

Let $R = f_{sim}(\mathcal{G})$. Instead of using the default summation as the aggregation function $f_{agg}$, we opt to normalize $R$ and take the negative entropy to capture the relative contributions of each node and graph filter. We define the aggregation function $f_{agg} = \left( f_{agg}^{(1)}, \ldots, f_{agg}^{(m)} \right)$, where

$$f_{agg}^{(i)}(R) = \sum_{v=1}^n \frac{R_{vi}}{\| R \|} \log \frac{R_{vi}}{\| R \|} \in \mathbb{R}, \quad i = 1, \ldots, m. \quad (4)$$

For the final predictor $f_{pred}$, we employ a single linear layer or MLP, preceded by batch normalization, but aim to use as little layers as possible to achieve desired accuracy. This setup facilitates optimization, encourages the model to learn a more effective set of graph filters, and helps prevent overly complex dependencies between them and final predictions. The graph filters $\mathcal{H}_1, \ldots, \mathcal{H}_m$, along with the parameters of the predictor function $f_{pred}$, are parameters of the network optimized during training using gradient descent. Figure 3 shows an overview of XGKN.

## 4.2 EXPERIMENTS

### 4.2.1 SETUP

In terms of datasets, metrics and thresholding, we follow the same setup described in Section 3.2.1.

**Hyperparameters** For GKN-specific hyperparameters, we do a grid search considering those best suited for GKNs models from Section 3. Searched features include: number of graph filters (4, 8 or 16), size of graph filters (6 or 8), dimension of the node feature encoder (16), radius of node-centered subgraphs (2 or 4) and their max size (10). We employ a single-layer classifier, preceded by batch normalization, to make the final prediction. We train XGKN for up to 1000 epochs using the Adam optimizer, with a learning rate of 0.01, a weight decay of 1e-4, and a batch size of 64.

### 4.2.2 RESULTS

Table 4 shows predictive accuracy and time needed to extract explanations using XGKN. Figure 4 shows achieved AIM metrics.

XGKN outperforms its predecessor, KerGNN, delivering superior results while requiring less time to extract explanations compared to other methods. It identifies more relevant concepts, as evidenced by higher scores in A2 and M1-M3. Additionally, XGKN demonstrates consistency across different datasets, with more balanced metric values. Unlike other methods, where strong performance in one often leads to significant declines in others, XGKN maintains a more stable performance. For easier model comparison, refer to Appendix A.3.

Table 4: XGKN performance

|  | Accuracy (%) | Time (s) |
|---|---|---|
| BA-2motifs | 99.4 | 0.028 |
| BAMultiShapes | 91.2 | 0.028 |
| MUTAG | 84.74 | 0.006 |
| PROTEINS | 73.31 | 0.031 |
| IMDB-B | 67.30 | 0.028 |
| IMDB-M | 46.40 | 0.063 |

## 5 CONCLUSIONS

We propose AIM, a set of 12 metrics for a comprehensive evaluation of XAI methods for GNNs, addressing not only their accuracy when ground truths are available but also assessing the reliability

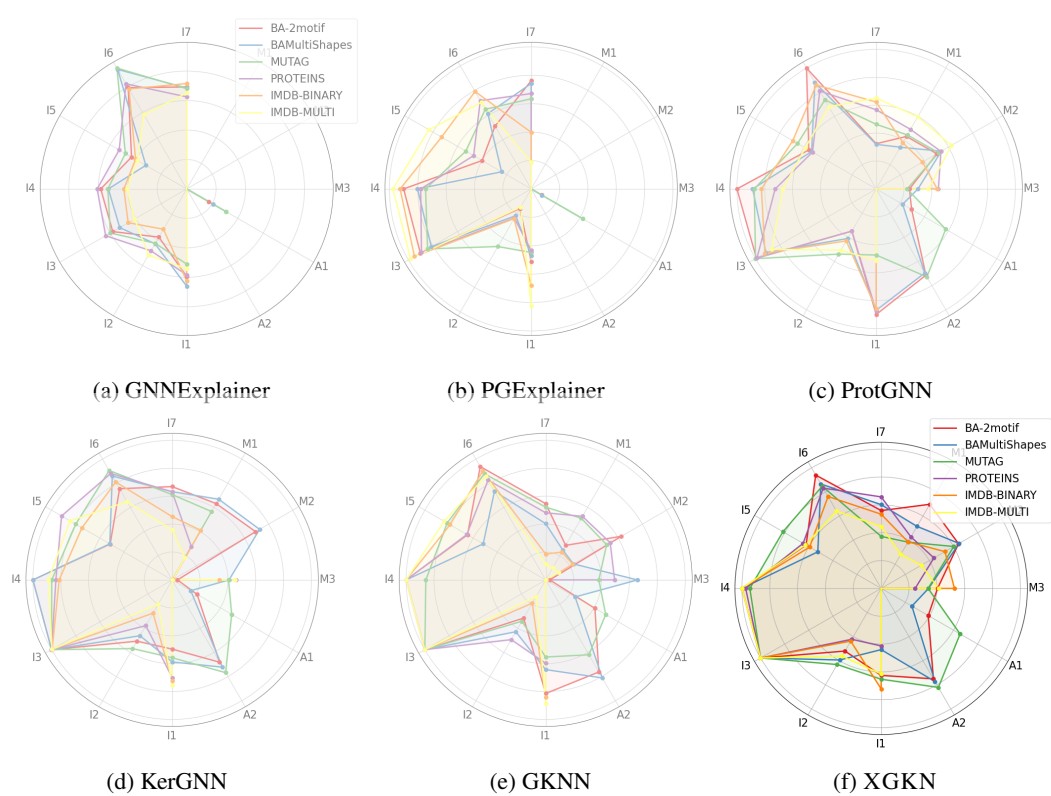

Figure 4: AIM metrics comparison for XGKN. Lower opacity figures same as Figure 2.

of both instance-level and model-level explanations. We define a way of extracting instance-level explanations from existing inherently interpretable GNNs, and demonstrate that AIM metrics effectively capture the strengths and limitations of XAI GNN methods. Based on observations, we propose a new model, called XGKN, which builds upon existing GKN principles while prioritizing XAI capabilities over just its accuracy.

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

# A APPENDIX

## A.1 HYPERPARAMETERS IN AIM EVALUATION

For AIM evaluation, certain perturbations are performed on input graphs, model's concept parameters are altered, new classifiers are trained. Here, we specify hyperparameters for these operations.

For I4, input graph perturbation is performed by altering node features with a probability of 0.05, where assigning each feature a randomly selected value from the dataset. For I5, perturbation is carried out by removing an existing edge with a probability of 0.005 and inserting a new node between two connected nodes with the same probability. For I6, a new classifier is trained using the hyperparameters from the GIN model employed for post-hoc explainers. For M1, concept modification involves changing the features of each node with a probability of 0.5, assigning random features from the dataset, and encoding them if required. For M2, concepts are perturbed by either adding a non-existent edge or removing an existing edge between node pairs, each with a probability of 0.25.

## A.2 THRESHOLDING

To distinguish between relevant and irrelevant parts of input graphs, it is necessary to apply a threshold to the importance maps of nodes or edges. Several techniques can be used for this purpose.

One straightforward approach is the top-$k$ method, where the $k$ nodes or edges with the highest importance scores are selected. Another technique involves setting the threshold based on percentiles. In this case, importance maps are normalized to sum to one, and for a given percentile $p \in [0, 1]$, a threshold is chosen such that the cumulative importance below it sums to $p$. Nodes or edges with importance scores above this threshold are considered relevant.

A method that avoids the need for hyperparameters like $k$ or $p$ is identifying the elbow point of the sorted importance scores, which serves as a natural threshold. We apply the approach based on calculating distance from a reference line.

Figure 5 presents the results of different thresholding techniques applied to various models on the BA-2motif dataset.

We observe correlations between different metrics. Improvement in one often comes at the cost of another; for example, increasing the size of the explanation (I7) improves the likelihood that the explanation will be classified in the same class as the original graph (I1). We see that AIM evaluation for some models depends more heavily on the thresholding technique (PGExplainer, ProtGNN, GKNN).

Ideally, importance maps should be constructed in a way that allows for clear distinction between relevant and irrelevant nodes, for instance, by leveraging elbow points, rather than relying on hyperparameters such as the target number of nodes or a percentile.

## A.3 SUPPLEMENTARY PLOTS FROM AIM EVALUATION

Figure 6 presents additional plots of the results shown in the main paper, aggregated based on the used dataset for easier comparison of the models.

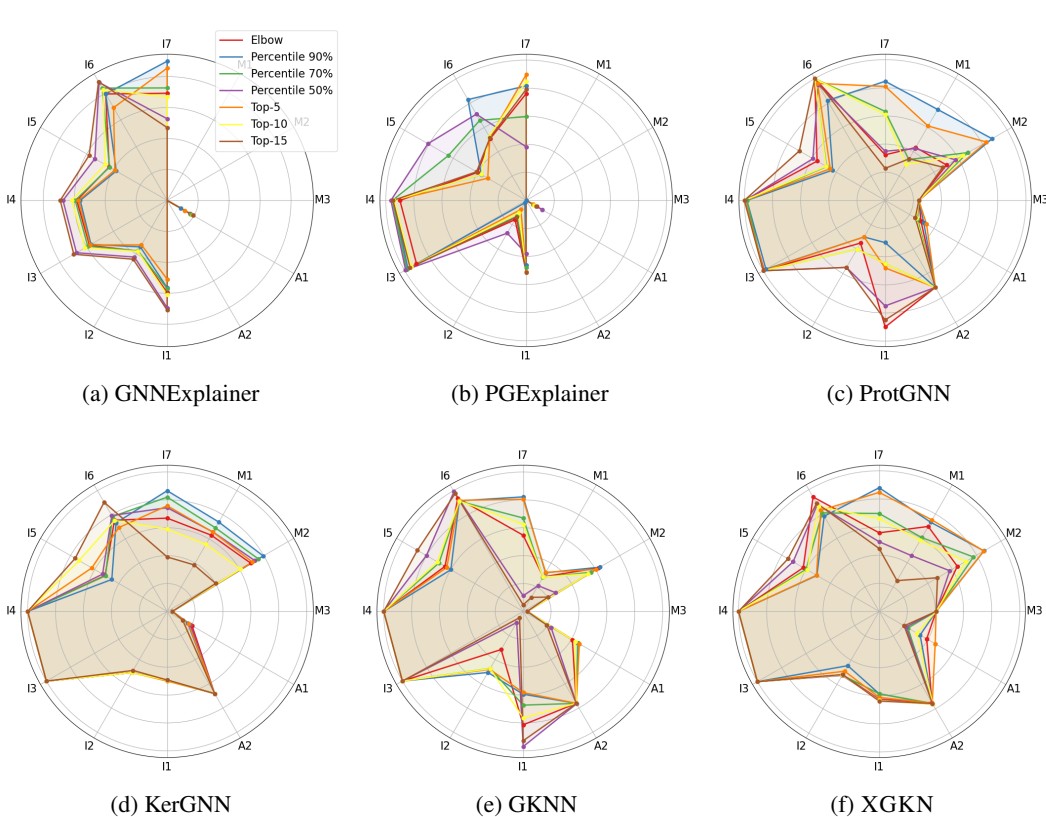

Figure 5: AIM metrics measured for different thresholding techniques and models on the BA-2motif dataset: 1) post-hoc explainers: GNN-Explainer, PGE-Explainer, 2) Prototypical Network: Prot-GNN, and 2) Graph Kernel Networks: KerGNN, GKNN and XGKN. Since PGExplainer produces importance maps for edges rather than nodes, for top-$k$ experiments, we select nodes from the top $\lceil k/2 \rceil$ edges to ensure a fair comparison. Note that the metrics have been adjusted such that higher values indicate better performance.

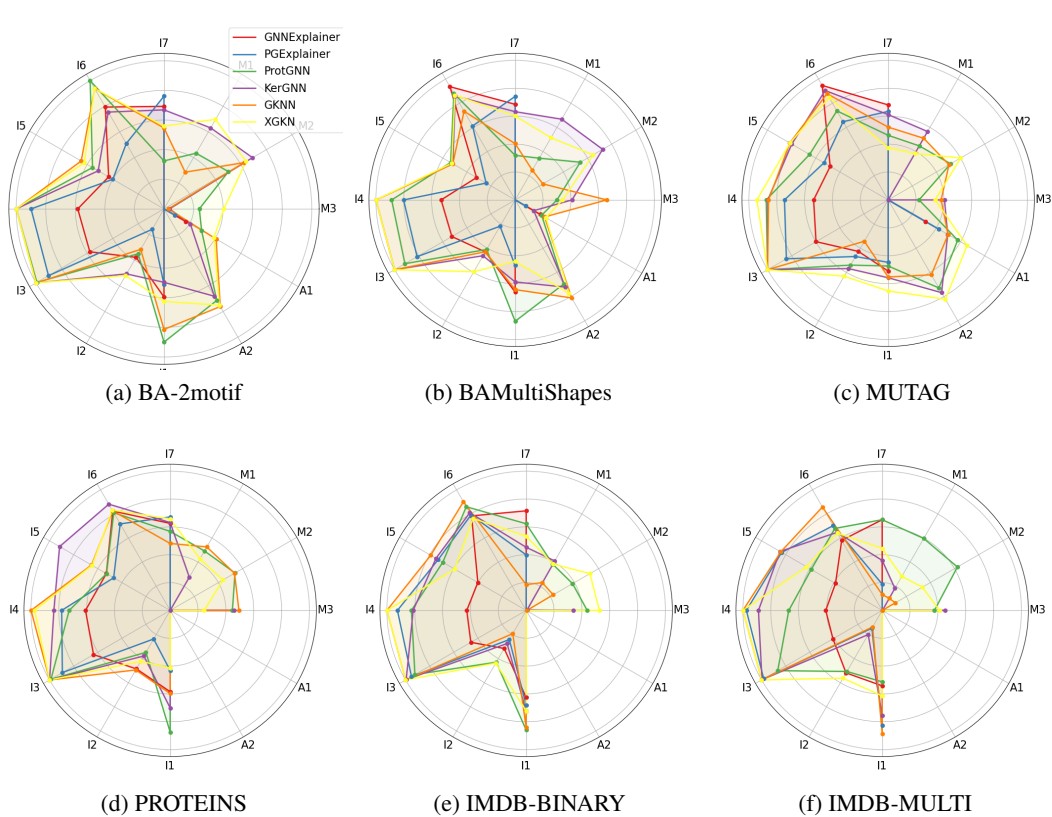

Figure 6: Comparison of AIM metrics achieved by different models, aggregated by datasets.

