# OpenReview forum: "AIMing for Explainability in GNNs"
_ICLR.cc/2025/Conference — Submitted to ICLR 2025_

### Official Review · Reviewer_SFvc · 2024-10-17

**Soundness:** 1
**Presentation:** 1
**Contribution:** 1
**Rating:** 3
**Confidence:** 5

**Summary:**

The authors propose the adoption of the Co-12 properties previously developed in explanations for images into the graph domain.
The authors also claim that explainability for graphs is a rather under-studied issue, and that lacks standardized evaluation metrics.
After having defined those evaluation metrics, the authors propose a way to extract local explanations from previous explainable models and propose a new variant improving on self-explainability.

**Strengths:**

- The overall quality of the writing is good
- Challenging common assumptions in the literature, such as evaluation metrics, is useful for the community
- The visualization of results across different metrics (Fig. 1) is nice

**Weaknesses:**

The paper **substantially lacks references to previous works**, and **many claims central to the proposed contribution are contradicted by the literature**. For example:

1. line 14-15: the claim is not supported, as plenty of approaches evaluate post-hoc methods with metrics not requiring ground truth [1,2,3]
2. line 15-16: the contrasting approach to post-hoc methods is referred to as ante-hoc methods. The indicated ante-hoc methods (prototypes and graph kernels) constitute only a very limited perspective on the current state of the art in ante-hoc methods [1,4].
3. line 37: the claim is not supported, see [1,2,3,4]. XAI for GNNs is actually a very active field of research
4. line 51: the claim is not supported, see my point 2
5. line 60-61: the claim is not supported, as plenty of metrics are available for graph explanations [1,2,3,4]

[1] Explaining the Explainers in Graph Neural Networks: a Comparative Study. Longa et al. 2024. ACM Comput. Surv.

[2] GraphFramEx: Towards Systematic Evaluation of Explainability Methods for Graph Neural Networks. Amara et al. 2022. LoG

[3] Evaluating explainability for graph neural networks. Agarwal et al. 2023. Sci Data

[4] A Survey on Explainability of Graph Neural Networks. Kakkad et al. 2023. LoG

**Questions:**

- Line 143 says "Explainers do not have to be deterministic". I wonder which kind of insights a human can extrapolate from a black-box when the explanation for a certain fixed output changes when the explanation is non-deterministic. Also, PGExplainer is indicated as a non-deterministic algorithm. However, once the explainer is trained, it is fully deterministic. Only the training of such an explainer is non-deterministic. This should be clarified in the text.

- The authors are proposing to evaluate graph explanation with a suite of 12 evaluation metrics. Such a high number of metrics, in my opinion, makes the assessment of different explainers difficult, making comparison hard.

- The proposed model is difficult to understand, and Figure 3 does not provide enough information. I would suggest annotating the figure with more detail and providing a more intuitive explanation.

- Some important implementation details for ensuring reproducibility are lacking, like how the Elbow method for obtaining explanation threshold is implemented. I suggest providing more details on this

---

> ### Author Response · Authors · 2024-11-25
>
> We appreciate the feedback and have revised the manuscript accordingly.
>
> > The paper substantially lacks references to previous works, and many claims central to the proposed contribution are contradicted by the literature.
>
> We have refined the introduction and related work section, considering the points raised by reviewers.
>
> > Line 143 says "Explainers do not have to be deterministic". I wonder which kind of insights a human can extrapolate from a black-box when the explanation for a certain fixed output changes when the explanation is non-deterministic. Also, PGExplainer is indicated as a non-deterministic algorithm. However, once the explainer is trained, it is fully deterministic. Only the training of such an explainer is non-deterministic. This should be clarified in the text.
>
> We added clarification to the manuscript.
>
> > The authors are proposing to evaluate graph explanation with a suite of 12 evaluation metrics. Such a high number of metrics, in my opinion, makes the assessment of different explainers difficult, making comparison hard.
>
> The broader set of metrics is intended to highlight the strengths and weaknesses of different methods by assessing multiple properties, which is challenging with aggregated metrics alone. Depending on the use case, these metrics could be selectively aggregated or only a subset could be used.
>
> > The proposed model is difficult to understand, and Figure 3 does not provide enough information. I would suggest annotating the figure with more detail and providing a more intuitive explanation.
>
> We have provided additional explanations for Figure 3 in the updated manuscript.
>
> > Some important implementation details for ensuring reproducibility are lacking, like how the Elbow method for obtaining explanation threshold is implemented. I suggest providing more details on this
>
> Code is shared to ensure reproducibility. We have also included a more precise information on the Elbow method in the Appendix.

---

> > ### Comment · Reviewer_SFvc · 2024-11-28
> > **Response by reviewer**
> >
> > I thank the authors for the clarifications.
> >
> >
> > > Depending on the use case, these metrics could be selectively aggregated or only a subset could be used.
> >
> > I think this is somewhat vague and does not give practical takeaways for practitioners on how to use and interpret the metrics.
> >
> >
> > > We have refined the introduction and related work section, considering the points raised by reviewers.
> >
> > I double-checked the revised paper, but it is unclear why the evaluation metrics available for post-hoc methods are unsuitable for ante-hoc methods. The authors do not provide a strong argument for motivating the use of the 12 new metrics, and also the experiments lack comparisons with previous metrics.

---

### Official Review · Reviewer_CoQQ · 2024-10-29

**Soundness:** 2
**Presentation:** 2
**Contribution:** 3
**Rating:** 5
**Confidence:** 4

**Summary:**

In this paper, the authors propose novel metrics for evaluating GNN explainer models, as well as a new interpretable GNN model based on KerGNN, called XGKN. The work is innovative and has great potential in the field of graph neural networks (GNNs). The authors have conducted extensive experiments to assess both their proposed metrics and the performance of their new model. However, there are several concerns that need to be addressed to improve the clarity and presentation of the paper.

**Strengths:**

The authors tackle an important problem in GNN interpretability and propose both a new model and metrics to evaluate it. The concept of XGKN and the proposed evaluation metrics are novel and relevant. Furthermore, the experimental evaluation is thorough, providing substantial support for the contributions.

**Weaknesses:**

1. Introduction Structure and Clarity:
   The logic in the introduction is somewhat unclear. The authors should clearly state their two main objectives: (1) to propose new evaluation metrics for GNN explainability, and (2) to introduce XGKN, a GNN model with inherent interpretability. After discussing the shortcomings of existing methods, the authors should introduce their model XGKN, explaining how it addresses the identified limitations. This should be followed by a clear explanation of the shortcomings of current evaluation metrics, leading to their motivation for proposing new metrics. Additionally, it would be helpful to move the related work to a separate section, so the introduction can focus more on the motivation and contributions. The current contribution section lacks clarity and should be more explicit about the novelty of the work.

2. Writing and Formatting Issues:
   There are numerous writing errors throughout the paper:
   - The caption for the tables is not formatted correctly. The captions should be placed above the tables.
   - In line 81, "GKN" should be "XGKN," as it refers to the authors' proposed model.
   - Line 140: A comma is missing after "as an explanation" and before "for the prediction."
   - Line 141: The correct notation should be "IoU(G1, G2)."
   - Line 142: The symbol "II" is introduced without explanation, and should be defined clearly when first mentioned.
   - Line 153: The formula should be written as "II(f(G \ h(G)) ≠ c)."
   - Lines 156 and 159 use "G'" to represent different meanings. This inconsistency is confusing, and different symbols should be used to avoid ambiguity. Furthermore, "G'" appears again in line 374 without proper explanation.
   - Lines 160 and 213: The "G" font is incorrect; it should be italicized to represent "graph."
   - Line 238: There is a missing "a" before "GIN."

3. Formula and Notation Issues:
   - Line 160: The explanation for contrastivity is not clear. Contrastivity measures the distinguishability between explanations for different predictions. The authors should clarify their reasoning behind the current formula and explain how it relates to the concept of contrastivity as typically understood in the field.
   - In lines 149 and 167, the definitions of A1 and A2 rely on the explanation of ground truth. However, for real-world datasets where ground truth is not available, it is unclear how this would be handled. The authors could propose alternative metrics for cases without ground truth or discuss how their method could be adapted for such scenarios.
   - Line 377: The meaning of "K" in the formula is not explained and should be clarified.

4. Figure and Table Organization:
   Figures 4 and Table 4 should be combined with Figure 2 and Table 3, respectively. This would allow for better comparison between the performance of the proposed method and existing methods, making it easier to assess improvements.

5. Overall Organization and Presentation:
   While the paper presents innovative ideas, the structure and presentation need significant improvement. It is recommended to first introduce the proposed method (XGKN), followed by the proposed metrics. In the experiments section, the authors should use their proposed metrics to evaluate both existing methods and their own. Additionally, the limitations of existing metrics should be clearly stated, and the authors should explain how their proposed metrics address these limitations.

**Questions:**

See in Weeknesses.

---

> ### Author Response · Authors · 2024-11-25
>
> Thank you for the feedback and for noting the corrections in notation and punctuation. We have updated the manuscript accordingly.
>
> > 1 Introduction Structure and Clarity: The logic in the introduction is somewhat unclear. The authors should clearly state their two main objectives: (1) to propose new evaluation metrics for GNN explainability, and (2) to introduce XGKN, a GNN model with inherent interpretability. After discussing the shortcomings of existing methods, the authors should introduce their model XGKN, explaining how it addresses the identified limitations. This should be followed by a clear explanation of the shortcomings of current evaluation metrics, leading to their motivation for proposing new metrics. Additionally, it would be helpful to move the related work to a separate section, so the introduction can focus more on the motivation and contributions. The current contribution section lacks clarity and should be more explicit about the novelty of the work
>
> We revised the introduction section to improve clarity.
>
> > Line 160: The explanation for contrastivity is not clear. Contrastivity measures the distinguishability between explanations for different predictions. The authors should clarify their reasoning behind the current formula and explain how it relates to the concept of contrastivity as typically understood in the field.
>
> We reference Contrastivity defined as in Co-12 [1], where "Contrastivity addresses the discriminativeness of explanations with respect to a ground-truth label or other target".
>
> > In lines 149 and 167, the definitions of A1 and A2 rely on the explanation of ground truth. However, for real-world datasets where ground truth is not available, it is unclear how this would be handled. The authors could propose alternative metrics for cases without ground truth or discuss how their method could be adapted for such scenarios.
>
> AIM metrics are proposed to assess models where ground truths are not available. But in case they are, it is reasonable to assess accuracy using them. Some metrics are not possible to evaluate (e.g. model-level explanations for considered post-hoc explainers), in that case they are skipped in evaluation (see Figure 2).
>
> > Figure and Table Organization: Figures 4 and Table 4 should be combined with Figure 2 and Table 3, respectively. This would allow for better comparison between the performance of the proposed method and existing methods, making it easier to assess improvements.
>
> Improvements were proposed based on results achieved in Figure 2 and Table 3. Combined results from figures are in Appendix. We have included earlier results into Figure 4 (dehighlighted) to make the comparison between models easier.
>
> > 5. Overall Organization and Presentation: While the paper presents innovative ideas, the structure and presentation need significant improvement. It is recommended to first introduce the proposed method (XGKN), followed by the proposed metrics. In the experiments section, the authors should use their proposed metrics to evaluate both existing methods and their own. Additionally, the limitations of existing metrics should be clearly stated, and the authors should explain how their proposed metrics address these limitations.
>
> The paper is organized to first establish the desired properties for explainability, then evaluate existing GKNs in relation to these properties, and finally propose improvements based on this analysis. Existing metrics mainly focus on fidelity (referred to as Completeness in our work and aligned with the Co-12 properties), but lack evaluation of other critical properties—a limitation we aim to address.
>
> [1] Meike Nauta et al., "From Anecdotal Evidence to Quantitative Evaluation Methods: A Systematic Review on Evaluating Explainable AI.", ACM Comput. Surv. 2023.

---

### Official Review · Reviewer_cxxy · 2024-10-30

**Soundness:** 2
**Presentation:** 2
**Contribution:** 2
**Rating:** 3
**Confidence:** 4

**Summary:**

The submission first proposes AIM metrics that cover Co-12 properties for evaluation of the Explainer of GNN Models. The authors then validate different existing GNN explainers with the proposed metrics. Moreover, the authors propose a new explainer approach based on Graph Kernel Networks (GKN). Basically, the idea of the proposed explainer is not very novel, that is a simple extention of existing GKN approaches. The thorough validation on different properties of explaination of an explainer of GNN is important.

**Strengths:**

1). The thorough validation on different properties of explaination of an explainer of GNN is important. Exisiting work does not discuss such different properties thoroughly. This is a good contribution of this submission.

2). The problem is important for XAI, espeicially for applications in important AI for sicience tasks, like drug discovery.

**Weaknesses:**

1). the idea of the proposed explainer is not very novel. The paper listed two other GKN approaches. On top of that, the new stuffs in this paper is not significant. Meanwhile, the performance improvement on AIM metrics over two other GKN approaches is not significant.

2). The authors do not consider the OOD issue of the metrics that are discussed in the following papers:
Cooperative explanations of graph neural networks. WSDM’23.
TOWARDS ROBUST FIDELITY FOR EVALUATING EXPLAINABILITY OF GRAPH NEURAL NETWORKS.  ICLR’24.

3). The motivation is not well-written on the improvement over existing GKN approaches.

**Questions:**

as mentioned in weakness.

---

> ### Author Response · Authors · 2024-11-25
>
> Thank you for your feedback. We have revised the manuscript and addressed your concerns in the responses below.
>
> > 1). the idea of the proposed explainer is not very novel. The paper listed two other GKN approaches. On top of that, the new stuffs in this paper is not significant. Meanwhile, the performance improvement on AIM metrics over two other GKN approaches is not significant.
>
> Our main goal is to propose a comprehensive set of metrics to evaluate XAI GNN approaches. Such framework evaluates desired properties, allows to determine shortcomings and improve methods. We showcase that by investigating GKNs. We note that improvements are not ground breaking, but that was not our objective.
>
> > 2). The authors do not consider the OOD issue of the metrics that are discussed in the following papers: Cooperative explanations of graph neural networks. WSDM’23. TOWARDS ROBUST FIDELITY FOR EVALUATING EXPLAINABILITY OF GRAPH NEURAL NETWORKS. ICLR’24.
>
> The claim that f(h(G)) is being applied ood when what is trained is f(G) is generally true, but for a range of approaches, particularly GKNNs and KerGNNS, the fact that we operate over subgraphs anyways is some sort of protection against complete ood.
> More generally, the question of whether an explanation---defined to be an 'importance' scored subgraph in the input---is necessary and sufficient to recover the intended prediction is quite reasonable. The main issue appears to be whether the architecture employed for f is amenable to this, and in certain cases, it is much less susceptible to ood than others.
>
> > 3). The motivation is not well-written on the improvement over existing GKN approaches.
>
> We present our motivation following an evaluation of existing GKNs, outlining our thought process and rationale for improvements.

---

> > ### Comment · Reviewer_cxxy · 2024-11-25
> > **Thanks for the response**
> >
> > Thanks for the response.

---

### Official Review · Reviewer_FTh2 · 2024-11-04

**Soundness:** 3
**Presentation:** 3
**Contribution:** 2
**Rating:** 3
**Confidence:** 4

**Summary:**

In this paper, the authors focus on the problem of GNN explainability;

- Contribution 1 -- a framework for graph explainability, with 12 properties

- Contribution 2 -- XGKN, a new explanable GNN model based on random walks

**Strengths:**

- This paper tackles an important problem of GNN explainability
- The proposed method XGKN seems to provide some benefits against KerGNN (though see Questions)

**Weaknesses:**

- It would be great if the authors put their work in the context of previous efforts, especially for C1
- Some questions about experiments

**Questions:**

1. In terms of contribution 1, there are previous efforts such as GraphFramEx that try to provide
some properties and eval metrics for graph explainability. Although I understand AIM is not
exactly the same, it would be great if the authors can elaborate on the differences and build their
insights over previous work instead of starting from scratch

2. Personally I found it rather hard to compare XGKN and KerGNN by cross ref Figure 2 and Table 3
with Figure 4 and Table 4 that are 3 pages apart. But my understanding is that:

- Table 3 v Table 4 -- out of 6 benchmarks, 3 KerGNN is better and 3 XGKN is better -- it would be
great if the authors can elaborate on why "XGKN outperforms its predecessor, KerGNN, delivering superior results"

- In terms of times to extract explanations -- the different is quite small and probably fall in the
regime of implementation details -- it would be great if the authors can elaborate on why this difference
is fundamental, rather than some implication of implementation decisions.

- I didn't find a good way to compare Figure 2 and Figure 4 clearly and understand it; given the authors still
have 1 page left, I would enrich 4.2.2 and really compare these two methods clearly, tease apart different
datasets.

---

> ### Author Response · Authors · 2024-11-25
>
> Thank you for your feedback regarding the paper. We have revised the manuscript accordingly and added further clarifications in the responses below.
>
> > In terms of contribution 1, there are previous efforts such as GraphFramEx that try to provide some properties and eval metrics for graph explainability. Although I understand AIM is not exactly the same, it would be great if the authors can elaborate on the differences and build their insights over previous work instead of starting from scratch
>
> The related work section has been expanded to include additional references to evaluation metrics proposed for post-hoc explainers such as GraphFramEx. We also emphasized the similarities between our metrics and those discussed in other works.
>
>
> > Personally I found it rather hard to compare XGKN and KerGNN by cross ref Figure 2 and Table 3 with Figure 4 and Table 4 that are 3 pages apart. But my understanding is that:
>
> > Table 3 v Table 4 -- out of 6 benchmarks, 3 KerGNN is better and 3 XGKN is better -- it would be great if the authors can elaborate on why "XGKN outperforms its predecessor, KerGNN, delivering superior results"
>
> Our aim was to first analyze existing methods, identify their limitations, and then introduce our model (XGKN) to address these gaps, hence the paper’s structure and the division of results across two tables and two figures. Accuracy levels across models are generally comparable, with notable outliers that indicate the relative difficulty of tasks for specific models (e.g., PROTEINS for ProtGNNs and IMDB for GKNNs). Since our primary focus is on interpretability, Figures 2 and 4 are most relevant and our comments on XGKN’s performance improvements refer to these results.
>
>
> > In terms of times to extract explanations -- the different is quite small and probably fall in the regime of implementation details -- it would be great if the authors can elaborate on why this difference is fundamental, rather than some implication of implementation decisions.
>
> For explanations extraction we use publicly available code with additionally needed code shared across models to achieve fair comparison. The goal was to highlight time needed to extract explanations from ProtGNN and explain that it is due to the sampling strategy that ProtGNN employs, which we do in section 3.2.2.
>
>
> > I didn't find a good way to compare Figure 2 and Figure 4 clearly and understand it; given the authors still have 1 page left, I would enrich 4.2.2 and really compare these two methods clearly, tease apart different datasets.
>
> We have updated Figure 4 to include prior results (dehighlighted) to make comparison easier. Nevertheless, Figure 6 in the Appendix provides results for all datasets, aggregated appropriately.

---

### Official Review · Reviewer_sYXV · 2024-11-05

**Soundness:** 2
**Presentation:** 3
**Contribution:** 2
**Rating:** 3
**Confidence:** 4

**Summary:**

The authors propose a set of metrics for measuring explainability of GNNs inspired by the Nauta et al. co-12 properties. They also develop a variant of graph-kernel based GNNs enhancing explainability properties.

**Strengths:**

The idea of evaluating explainability from many different perspectives is (not novel but) sensible, and it's application to GNN explainability is potentially interesting.

The proposed approach seem to strike a good balance between different metrics.

**Weaknesses:**

My impression is that the authors made an attempt to adapt the co-12 properties from (Nauta et al., 2023), but partly failed to account for the specificities of of networked data and GNNs, and overlooked a number of formalization efforts already available in the literature.

Completeness (without): I(f(G\f(h(G)) != c) the formula is wrong, I
guess it should be I(f(G \ h(G)) != c). Also, this metric exists and
is called fidelity- (or sufficiency).

Consistency: this is poorly formalized, as IoU(h(G),h(G)) is 1 by
construction. You should clarify (and formally write) that this is
computed by picking explanations from a distribution (I guess).

Continuity: While this is sensible with low-level data (like pixels),
it's not necessarily sensible with discrete structures like graphs,
where the the ground truth could be a motif, and removing elements of
the motif could end up changing the label of the graph. Indeed,
measures like explanation fidelity (especially robust fidelity+),
encode the fact that removing elements that are part of the
explanation should have an impact on the prediction of the model (and
thus, on the resulting post-hoc explanation).

Contrastivity: the definition doesn't seem to fit the description in Table 1

For model-level metrics, correctness seems to be closer to a
(model-level) definition of robust fidelity+, but in this set of
metrics is somehow in contradiction with consistency in the
instance-level metrics. Also

M3: Compactness - I am not sure this is about compactness, I would
rather talk about non-redundancy.

Talking about interpretable GNNs, the authors only focus on
prototype-based and kernel-based GNNs. However, the literature on
interpretable GNNs is much richer. For instance, all methods based on
attention are missing, e.g.:

Siqi Miao, Mia Liu, and Pan Li. Interpretable and generalizable graph learning via stochastic attention mechanism. ICML 2022.

Chris Lin, Gerald J Sun, Krishna C Bulusu, Jonathan R Dry, and Marylens Hernandez. Graph neural networks including sparse interpretability. arXiv, 2020.

Giuseppe Serra and Mathias Niepert. Learning to explain graph neural networks. arXiv, 2022.

This limits the relevance of the experimental evaluation. Additionally, only two explainers are evaluated, while  plenty of (more recent) explainers have been proposed in the literature.

Finally, the authors propose a SHAP-based approach to extract explanations from interpretable GNNs, which is fine. But post-hoc explainers can still be applied to interpretable GNNs, so it is not true that no  solution exists. The authors should show that their approach is better in extracting explanations from interpretable GNNs.

**Questions:**

Can you comment on my concerns about (some of) the properties you propose?

Can you comment on my concerns about the insufficient treatment of interpretable GNNs?

Can you comment on my concerns about the insufficient treatment of GNN explainers?

---

> ### Author Response · Authors · 2024-11-25
>
> Thank you for your feedback regarding the paper, we have updated the manuscript accordingly.
>
>
> > Completeness (without): I(f(G\f(h(G)) != c) the formula is wrong, I guess it should be I(f(G \ h(G)) != c). Also, this metric exists and is called fidelity- (or sufficiency).
>
> We corrected typo in the formula and added reference to fidelity. We follow the Co-12 [1] naming convention to unify evaluation framework for different modalities, e.g. our Completeness definition is similar to the accuracy-based fidelity [2].
>
>
> > Consistency: this is poorly formalized, as IoU(h(G),h(G)) is 1 by construction. You should clarify (and formally write) that this is computed by picking explanations from a distribution (I guess).
>
> We corrected the formula.
>
>
> > Continuity: While this is sensible with low-level data (like pixels), it's not necessarily sensible with discrete structures like graphs, where the the ground truth could be a motif, and removing elements of the motif could end up changing the label of the graph. Indeed, measures like explanation fidelity (especially robust fidelity+), encode the fact that removing elements that are part of the explanation should have an impact on the prediction of the model (and thus, on the resulting post-hoc explanation).
>
> The goal is to measure if background noise affect explanations. We have modified the noising process to generate the perturbed graph such that it excludes parts of graph that are part of initial graph explanations. This metric is related to robustness [3], but fundamentally distinct. Our experiments using this modified noising process appear to indicate little difference from the original results, although we appreciate that there could be more complex situations where this distinction is relevant.
>
>
> > Contrastivity: the definition doesn't seem to fit the description in Table 1
>
> We have updated the definition, which should hopefully clarify the confusion.
>
>
> > For model-level metrics, correctness seems to be closer to a (model-level) definition of robust fidelity+, but in this set of metrics is somehow in contradiction with consistency in the instance-level metrics.
>
> Our Model-level Correctness assesses changes in explanations when interpretable model parts (prototypes or graph kernels that guide explanation extraction) are modified. We do not see how this relates to robust fidelity+ (assuming that review refers to [2]) or conflicts with instance-level consistency.
>
>
> > M3: Compactness - I am not sure this is about compactness, I would rather talk about non-redundancy.
>
> Compactness in [1] covers size but also redundancy. We adopt this terminology to remain faithful to the Co-12 nomenclature.
>
>
> > Talking about interpretable GNNs, the authors only focus on prototype-based and kernel-based GNNs. However, the literature on interpretable GNNs is much richer. For instance, all methods based on attention are missing
>
> Attention-based methods are now included in the related work section.
>
>
> > Additionally, only two explainers are evaluated, while plenty of (more recent) explainers have been proposed in the literature.
>
> Our focus is on interpretable GNNs that provide model-level explanations, specifically graph kernel and prototype networks. To provide context and demonstrate the versatility of our approach, we also include two popular post-hoc explainers. Our goal was not to determine the best existing explainer but to establish a framework, detailing desired properties which should be optimized in future XAI GNN models. We focus on enhancing graph kernel networks, recognizing their potential for XAI. To this end, we identify desirable properties for XAI, outline graph kernel networks’ limitations, compare them to prototype networks (since they are most obvious alternative), and suggest improvements.
>
>
> > Finally, the authors propose a SHAP-based approach to extract explanations from interpretable GNNs, which is fine. But post-hoc explainers can still be applied to interpretable GNNs, so it is not true that no solution exists. The authors should show that their approach is better in extracting explanations from interpretable GNNs.
>
> While post-hoc explainers can extract explanations from interpretable GNNs, this setup complicates evaluation; it’s unclear whether limitations stem from the GNN’s interpretability or the explainer’s efficacy. For clarity, we use SHAP as a more straightforward method. Additionally, we show that explanations from post-hoc baselines are suboptimal, suggesting caution in relying solely on these approaches.
>
>
>
> [1] Meike Nauta et al., "From Anecdotal Evidence to Quantitative Evaluation Methods: A Systematic Review on Evaluating Explainable AI.", ACM Comput. Surv. 2023.
>
> [2] Xu Zheng et al., "Towards Robust Fidelity for Evaluating Explainability of Graph Neural Networks," ICLR 2024.
>
> [3] Mohit Bajaj, Lingyang Chu, Zi Yu Xue, Jian Pei, Lanjun Wang, Peter Cho-Ho Lam, and Yong Zhang. Robust counterfactual explanations on graph neural networks.

---

> > ### Comment · Reviewer_sYXV · 2024-12-02
> > **thanks for the answer**
> >
> > I would like to thank the authors for their clarification and for correcting errors and inaccuracies. I still think that the manuscript lacks in novelty (wrt existing work on explainability metrics) and generality. The focus on graph kernel and prototype networks seems to narrow (especially considering the generality of the title) and I disagree on the argument for dismissing the need for comparison of the SHAP-based approach with post-hoc explainers.

---

### Meta-Review · Area_Chair_LcHy · 2024-12-17

**Metareview:**

The authors propose a set of metrics inspired by the Co-12 properties (Nauta et al.) to evaluate the explainability of Graph Neural Networks (GNNs). They also introduce XGKN, a novel explainable GNN model, which is a variant of graph-kernel-based GNNs aimed at improving explainability. The paper claims to address shortcomings in current GNN explainability evaluation and models, providing both new evaluation tools and an enhanced method.

Strengths

+ The paper addresses the important and underexplored issue of GNN explainability, providing potentially useful metrics for systematic evaluation.
+ The proposed framework evaluates explainability across multiple properties, which is a comprehensive approach.
+ The authors provide thorough experimental validation for their metrics and model.
+ The visualization of results across different metrics is well-executed and informative.

Weaknesses

+ The proposed metrics partially fail to account for the specific nature of GNNs and graph-structured data, with several formalization issues noted (e.g., completeness, consistency, and continuity).
+ The paper lacks proper positioning within the existing literature on interpretable GNNs and explainers, missing related work and underexploring attention-based models and other approaches.
+ The novelty of the proposed XGKN model is limited, as it is seen as a minor extension of existing GKN methods with marginal performance improvements.
+ The experimental comparison with existing methods is insufficiently clear and lacks rigorous justification, particularly regarding performance claims and the relevance of metrics.
+ Writing and presentation issues (unclear introduction logic, ambiguous formulas, notation inconsistencies, and figure/table organization) further hinder clarity.
+ Implementation details necessary for reproducibility, such as thresholding techniques, are not well-described.

Some concerns have been addressed by the authors during the rebuttal period.

**Additional Comments On Reviewer Discussion:**

This paper received five negative ratings. Ever after discussion, reviewers seem to agree that this paper still has major issues that need to be improved, including novelty and motivation of the proposed new metrics, etc.

---

### Decision · Program_Chairs · 2025-01-22

Reject